# The Impact of the Quality of Logistics Services in E-Commerce on the Satisfaction and Loyalty of Generation Z Customers

Anh Duc Do *, Van Loi Ta, Phuong Thao Bui, Ngoc Thuy Do, Quynh Tho Dong and Hoai Thu Lam

School of Trade and International Economics, National Economics University, 207 Giai Phong Road, Hanoi 11616, Vietnam; loitv@neu.edu.vn (V.L.T.); 11203602@st.neu.edu.vn (P.T.B.); 11207099@st.neu.edu.vn (N.T.D.); 11207027@st.neu.edu.vn (Q.T.D.); 11207037@st.neu.edu.vn (H.T.L.)
* Correspondence: ducda@neu.edu.vn

**Abstract:** With the boom in e-commerce activities in Vietnam, the market size is expected to reach USD 52 billion by 2025, showing that e-commerce is a highly potential market. This also means that the level of competition between businesses is extremely fierce, so it requires optimization in each activity, especially e-logistics, to ensure smoothness, accuracy, and safety in distributing goods to consumers. Therefore, this study focuses on determining which factors of e-logistics activities affect the satisfaction and loyalty of Generation Z customers and their influence. The team collected opinions from 510 customers who had purchased goods through an e-commerce platform and then analyzed them using Smart-PLS3. The results show that delivery time is the most critical factor determining customer satisfaction, while the availability of goods is the factor that contributes the most to the loyalty of Generation Z customers. In the context of e-commerce development in Vietnam, the research has contributed to business enterprises' scale of e-logistics service quality and assessed the importance of each factor so that enterprises can base on that to evaluate their service quality and improve satisfaction, loyalty to customers, and competitiveness.

**Keywords:** quality of e-logistics services; satisfaction; loyalty; generation Z customers

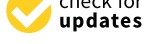



## 1. Introduction

Logistics is one of the most important means to improve the efficiency of the flow of materials, thereby reducing the distribution costs of the business. In addition to price, quality is also a factor creating competitive advantages which makes quality management one of the most important steps to achieve profit goals for commercial enterprises [1]. For quality management in general, it is impossible to ignore logistics service quality management. The research team led by Stank has shown a positive relationship between logistics service quality and customer satisfaction in the fast-food industry [2].

The trend of converting consumers' behavior from direct to online shopping is increasingly popular, most clearly shown during the COVID-19 epidemic in 2020. In Vietnam, this market is estimated at over USD 20 billion in 2023 with an average growth rate of 29% in the period 2020–2025 according to forecasts of Google (Mountain View, CA, USA), Temasek (Singapore) and Bain & Company (Boston, MA, USA). Specifically, the Generation Z customer group is considered to have the highest online purchasing power because by 2025, Generation Z will account for about 25% of the national workforce, and equivalent to about 15 million VND potential consumers. Specifically, in Hanoi—the capital of Vietnam—88% of Generation Z people have online shopping activities [3]. Therefore, optimizing e-logistics activities to meet the needs of this customer group is an important mission to complete.

Many studies have explored the positive influence of e-LSQ on customer satisfaction in the purchase process and customer loyalty [4,5]. Moreover, many other studies also show a positive relationship between customer satisfaction and loyalty [6,7]. However, research conducted by Balabanis's research group [8] shows that the basic factors leading to

purchase satisfaction do not have a direct impact on customer retention. Furthermore, when researching loyalty, some authors also argue that sometimes loyalty is just fake loyalty, that is, customers may continue to buy from that seller, but not because they feel satisfied, but for some other reasons such as: no other choice, high switching costs (the cost that the customer must pay when changing sellers or suppliers) [9]. In short, while customers are satisfied with the quality of service, they are not necessarily loyal to that online seller.

Therefore, the objective of this study is to analyze the mutual influence of the factors of e-logistics service quality with the satisfaction and loyalty of Generation Z customers in Hanoi; this study develops a scale to assess e-logistics service quality affecting satisfaction and loyalty of Generation Z customers. It adds new factors like payment methods tailored to online shopping and Generation Z. The findings help online sellers improve logistics services using quantified criteria to boost quality, satisfaction, and loyalty, particularly among Generation Z. Sellers need distinct brands and visual information to increase credibility and trust.

## 2. Literature Review

### 2.1. E-Logistics and E-Logistics Service Quality

In general, logistics is often approached from two perspectives. Firstly, logistics is the science of organizing and managing the distribution of goods and services from the pre-production stage until the goods reach the consumer. According to ESCAP [10], logistics is the management of the process of moving materials from warehousing to manufacturing into the final product to distribute to the consumers per their request. Secondly, logistics is also approached from the perspective of a service associated with the distribution and circulation process. In Vietnam, the official concept of logistics services was introduced for the first time in the 2005 Commercial Law of Vietnam (Article 233). "Logistic services are commercial activities whereby traders organize the performance of one or many jobs including reception, transportation, warehousing, yard storage of cargoes, completion of customs procedures and other formalities and paperwork, provision of consultancy to customers, services of packaging, marking, delivery of goods, or other services related to goods according to agreements with customers in order to enjoy service charges".

The concept of e-logistics was first introduced by Bayles [8]; according to the author, e-logistics is the application of logistics to conduct business in the electronic environment through the Internet. E-logistics is defined as the application of Internet-based technologies to support purchasing, storage, transportation, and enabling distribution through route optimization and inventory tracking [11]. By definition, e-logistics is the result of bringing e-commerce into logistics.

Meanwhile, the study of Han-ping [12] points out the fundamental differences between traditional logistics and e-logistics: while the distribution model of traditional logistics is "supply-driven push", e-logistics is "demand-driven pull", which shows that e-logistics is more flexible and responds better to market needs.

Based on the study on LSQ, e-LSQs are elements in the LSQ model operating in the online environment [4,13,14]. From the supplier's perspective, e-LSQ is measured by the ability to fulfill customer orders, i.e., the ability to fulfill orders and satisfy customers in electronic marketplaces [15,16].

Thus, it can be understood that the quality of logistics services in e-commerce represents the ability to satisfy customer needs during the shopping process in the e-commerce environment.

### 2.2. Customer Satisfaction

Customer satisfaction is a customer's overall or comprehensive assessment of the degree to which product or service performance matches expectations [17]. It can be perceived as a cumulative rating based on the total experience of purchasing and using a good or service over time [18–20]. Satisfaction is merely considered the first step in

improving customer retention and may not always positively affect the repeat purchase rate of customers in all cases.

A substantial number of findings strongly support the view that improving logistics service quality can increase customer satisfaction [21]. Elements of logistics service quality have been shown to have a positive relationship with customer satisfaction [14,22].

### 2.3. Customer Loyalty

Previous studies have typically identified one of two different approaches to the definition of customer loyalty. According to the first view, loyalty is simply a word to describe the ability to retain customers "A customer who continues to buy is a loyal customer" [23] (p. 218). The second view is that feelings and emotions influence customer loyalty; it develops on the behavioral aspect that implies repeat purchases or consumption stemming from attitude [24–26]. These views can be connected through the definition "Service loyalty is the degree to which a customer exhibits repeat purchasing behavior from a service provider, possesses a positive attitudinal disposition toward the provider, and considers using only this provider when a need for this service arises." [27] (p. 173).

Customer loyalty is conceptualized as having both behavioral and cognitive components [17,28]. Repurchasing behavior is derived from positive reviews of products/services provided, often used as a measure of customer loyalty [29]. However, repurchasing behavior can stem from the long-term desire of customers to maintain an important, valuable relationship with a supplier [30,31].

Griffis [5] showed that e-LSQ affects the recommendation of the product to new customers by old customers. Specifically, order fulfillment performance plays an important role in encouraging customers to recommend products to subsequent customers.

### 3. Research Hypotheses

### 3.1. The Relationship between Customer Satisfaction and Loyalty

Findings from previous studies have demonstrated that there exists a strong relationship between customer satisfaction and loyalty [14,17,18,21]. The factors of logistics services related to product availability, product status, etc. are believed to have a positive influence on customer satisfaction [21]. At the same time, studies in the context of online B2C also show a positive relationship between the factors of logistics services and customer loyalty [4,5,32].

However, contrary to the views of most of the above studies [33], in the online sales environment, even though customers are satisfied with the quality of the service quality, they may not be loyal to that online retailer. Thus, previous studies have shown a link between customer satisfaction and loyalty, but the results of these studies are not consistent. Consequently, to test the relationship with the group, the authors proposed the following hypothesis:

**H1:** *Satisfaction has a positive impact on Generation Z customer loyalty.*

### 3.2. Delivery Time

There have been many studies [13,14,34,35] showing the importance of delivery time or timeliness to customer satisfaction. "Timeliness" is a tool to measure customer loyalty and repurchase behavior [4,32]. Many Generation Z customers feel the need to buy and receive goods in the shortest time, especially when they are in urgent need of them for their activities that require said goods. Therefore, the authors believe that being able to choose a company and type of transportation (fast delivery within 2 h, express delivery) will have a positive impact on Generation Z customers and stimulate their purchases.

Based on the research results of the above scholars, in order to test the effect of delivery time on customer satisfaction and loyalty, the authors proposed the following two hypotheses:

**H2a:** *Delivery time has a positive impact on Generation Z customer satisfaction.*

**H2b:** *Delivery time has a positive impact on Generation Z customer loyalty.*

### 3.3. Availability

In the study of the Mentzer research group [16], this factor is studied under the name of "order release quantities". According to this study, based on several different factors, each organization can only have a certain amount of stock available, so it can only accept a few orders within its capacity. However, customers are only most satisfied when they can buy the right amount of goods they want. Therefore, the availability of goods has long been considered an important component of the LSQ [36].

In the study of omnichannel, the authors both pointed out that the availability of goods is one of the factors that directly affect both customer satisfaction and loyalty [37,38].

When conducting research with Generation Z customers, the authors found that pre-ordering (making an order for an unreleased item) is quite common. Limited edition products or custom-made products at the request of buyers are what the young Generation Z wants to own. Based on the above research and arguments, the following hypotheses were proposed:

**H3a:** *Availability has a positive impact on Generation Z customer satisfaction.*

**H3b:** *Availability has a positive impact on Generation Z customer loyalty.*

### 3.4. Information Quality

Information quality refers to the customer's perception of the information provided by the supplier regarding the products that the customer wants to inquire about. If this amount of information is available and the quality of the product based on that information is in line with the customer's needs, it can help the customer to make a purchase decision more quickly [14]. Especially with online shopping, customers are not able to touch and feel the product before purchasing it. Therefore, e-retailers should provide full information from specifications, origin, images, etc., about products to customers to limit confusion and misinformation. This minimizes the cost of reverse logistics to handle after-sales problems. This also makes customers feel more secure about their choice [22,39]. The authors therefore proposed the hypothesis:

**H4a:** *Information quality has a positive impact on Generation Z customer satisfaction.*

**H4b:** *Information quality has a positive impact on Generation Z customer loyalty.*

### 3.5. Product Quality and Condition

Much research points out that product quality refers to how well a product performs [14,34]. In an article, a research team led by Saura [22] also emphasized that product quality is an important aspect of logistics services in creating customer satisfaction. Damaged or defective products that leave customers unsatisfied can lead to customers not receiving the item, returning the item, or canceling the order. This causes great damage to the cost and the seller's reputation in the eyes of customers. The authors therefore proposed the following two hypotheses:

**H5a:** *Product quality and condition has a positive impact on Generation Z customer satisfaction.*

**H5b:** *Product quality and condition has a positive impact on Generation Z customer loyalty.*

### 3.6. Reverse Logistics

Reverse logistics refers to the process by which the product is returned by the consumer to a retailer or supplier for repair, resale, or recycling [40]. Research on returning goods from

the perspective of the seller found that the speed of processing returned goods is directly proportional to the retention rate, frequency, and quantity of purchases of customers [5]. Many sellers and e-commerce platforms support the return and exchange of goods, but the policies related to return and exchange are still inadequate. For example, employees may arrive late to collect goods, resulting in the buyer having to return to the post office so as not to miss the time limit for exchange and return. In addition, the buyer must properly package the goods to ensure that the goods are not damaged during transportation to the seller's shop. The rules for returns are sometimes so complicated that the customer feels that the seller is trying to make it difficult for them. Therefore, the authors have included the issue of the return and exchange process in the research paper.

**H6a:** *Reverse logistics has a positive impact on Generation Z customer satisfaction.*

**H6b:** *Reverse logistics has a positive impact on Generation Z customer loyalty.*

### 3.7. Customer Care

Customers' perception of service quality is closely related to the service delivery process, which itself is related to the relationship between customers and salespeople [39]. Therefore, the quality of customer care is an important aspect of the seller–buyer relationship [41,42]. Based on this theoretical foundation, customers will be interested in whether the seller is understanding and empathetic to their situation, actively helping them to solve their problems or not [14].

In Vietnam, with today's large number of online orders, delivery drivers have become busier than before. There have been cases where the delivery drivers had a poor attitude towards buyers, leaving a bad impression in the eyes of consumers. Therefore, contacting a delivery driver with a polite, gentle, and friendly attitude will leave buyers with a good impression. From the research results of the scholars and the above arguments, the authors proposed the following two hypotheses:

**H7a:** *Customer care has a positive impact on Generation Z customer satisfaction.*

**H7b:** *Customer care has a positive impact on Generation Z customer loyalty.*

### 3.8. Shipping Costs

In research in Vietnam, cost is conceptualized as a separate aspect, the third factor of logistics service performance, separate and distinct from the operational and relational components of service [21]. Based on the research results of [21,43], the author found a link between shipping costs and customer satisfaction. Therefore, to test the existence of the relationship between these factors, these hypotheses were proposed:

**H8a:** *Shipping costs have a positive impact on Generation Z customer satisfaction.*

**H8b:** *Shipping costs have a positive impact on Generation Z customer loyalty.*

### 3.9. Payment Method

Worku [44] pointed out convenience and ease of use as two of the many characteristics of electronic payment methods. Electronic payment allows online sellers to sell goods anytime, anywhere, and save the cost of printing invoices; moreover, consumers will have access to the global market.

In Vietnam, the goal is that by 2025, non-cash payments in e-commerce will reach 50% and by 2030, the value of the digital economy will account for about 20% of Vietnam's GDP. This shows the state's efforts in promoting online shopping and payment.

The authors believe that the payment method is one of the important factors creating e-LSQ, especially strongly affecting customer satisfaction and loyalty of Generation Z—the

generation where efficiency and convenience are brought to the fore. To test the relationship between this factor with customer satisfaction and loyalty, the authors proposed the following two hypotheses:

**H9a:** *Payment method has a positive impact on Generation Z customer satisfaction.*

**H9b:** *Payment method has a positive impact on Generation Z customer loyalty.*

Inheriting some factors in the SERVQUAL model [43], the traditional LSQ evaluation model [13,14,22], the multi-channel LSQ evaluation model [37,38], the research model [39] and adding the "Payment method" factor, the authors build the expected research model as follows. Figure 1 shows the research framework for this study. The questionaires of this study presented in Appendix A.

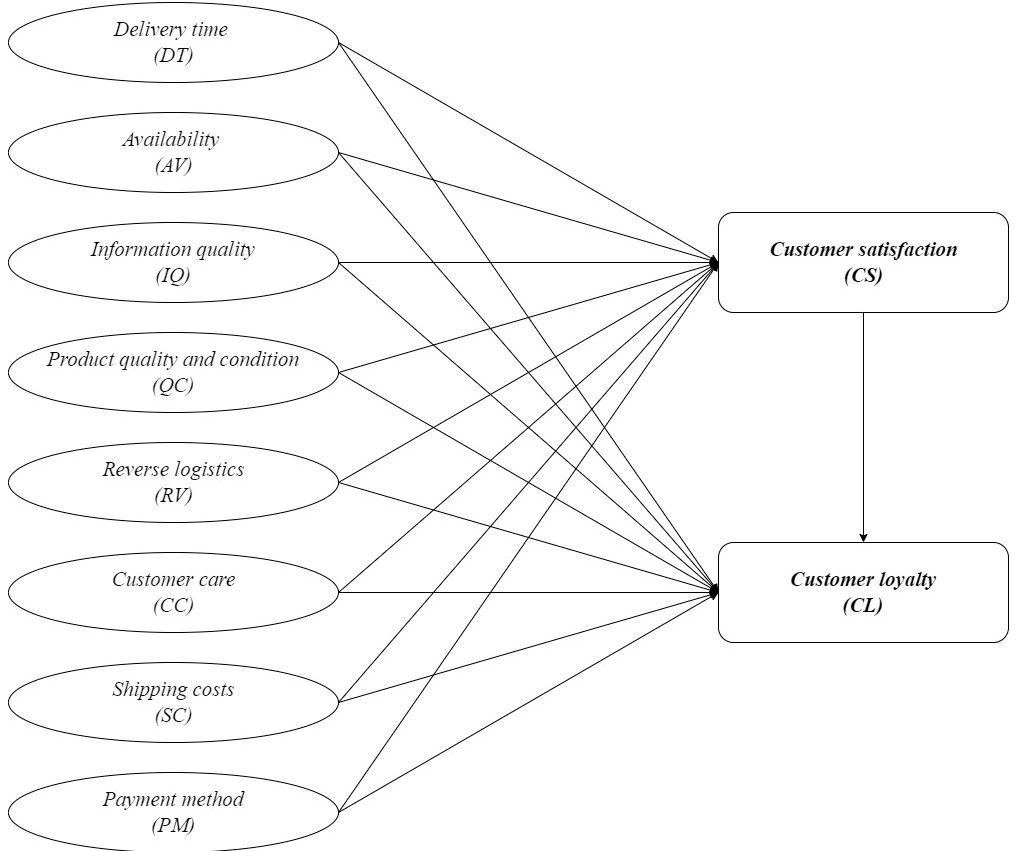

**Figure 1.** Research framework.

## 4. Research Methodology

### 4.1. Research Design

The research scale is built based on previous studies [14,21,35,37,39,45–47]. Furthermore, the authors develop several additional factors and observed variables so that the new scale is suitable for the research context in Hanoi, Vietnam and the research object is the Generation Z online consumer.

### 4.2. Data Analysis Techniques

#### 4.2.1. Qualitative

The authors conducted in-depth interviews with 10 Generation Z customers who have shopped online and experienced e-logistics services to determine the influence of e-logistics service quality on their satisfaction and loyalty. The main purpose is to modify the factors to fit the object, which is Generation Z online consumers in Hanoi.

### 4.2.2. Quantitative

The group conducted a preliminary quantitative study with 50 Generation Z customers in Hanoi city to test the scale obtained from qualitative interview results. The questionnaire uses a 5-point Likert scale (Strongly Disagree–Strongly Agree). Collected data will be checked and analyzed using Smart-PLS 3 software to produce the final questionnaire.

Sample sizes: The minimum sample will be 10 times the largest number of formative indicators used to measure a single construct [48]. In this research scale, the maximum number of indicators for any one factor is 7, so the sample size in this study is 70. The actual number of samples obtained was 510 samples by convenience sampling method.

After collecting the data, the team conducts synthesis and analysis. As the first step, the research team examined the average difference om the factors measuring logistics service quality between different income and age groups. Then, analyze the measurement model in the following order: verify the quality of the observed variable through the load factor, assess the reliability of the scale, and conduct the discrimination test. Finally, the evaluation of the linear structural model includes evaluation of multicollinearity using VIF, assessing the impact of the relationship, evaluating the explanatory level of the independent variable for the dependent variable through adjusted $R^2$, and evaluating the importance of the independent variables in the model by $f^2$ and evaluate the predictive power out of the sample through $Q^2$.

A total of 510 people responded to the survey. Specifically, in terms of gender, the number of women is 335 (65.7%), the number of men is 167 (32.7%) and the rest are other genders (1.6%). In terms of education level, the largest number of people are 324 people who have achieved a university degree, including those who have graduated and are currently studying (63.5%), followed by the number of people who have achieved the following level University such as graduate, master, doctorate (18.4%), who have or are currently studying college (13.3%) and the lowest is high school graduate (4.7%). Regarding income, there were 251 participants with income from over 10 million VND to 15 million VND, accounting for the majority (49.2%) and, the lowest was those with income below 5 million VND (3.9%). Table 1 describes demographic statistics by number and corresponding percentage.

**Table 1.** Descriptive statistics of participants' demographics.

| Criteria | Quantity | Percent (%) |
|---|---|---|
| Gender | | |
| Male | 167 | 32.7 |
| Female | 335 | 65.7 |
| Different | 8 | 1.6 |
| Education level | | |
| High School Graduate | 24 | 4.7 |
| College | 68 | 13.3 |
| University | 324 | 63.5 |
| Post-Graduate | 94 | 18.4 |
| Average Monthly Income (VND) | | |
| Under 5 million | 20 | 3.9 |
| From over 5 to 10 million | 73 | 14.3 |
| From over 10 to 15 million | 251 | 49.2 |
| From over 15 to 20 million | 141 | 27.6 |
| Over 20 million | 25 | 4.9 |

## 5. Research Results

### *5.1. T-Test*

From the results of the *t*-test, there are differences in most of the e-logistics service quality evaluation criteria except reverse logistics in different income groups, while different genders showed different differences in all criteria.

### *5.2. Measurement Proposed Research Model Assessment*

5.2.1. Quality of Observed Variables

To evaluate the quality of the observed variable (Table 2), it is necessary to rely on the outer loading coefficient; the observed variable has quality when the outer loading > 0.708 [48]. This means that the latent variable explained at least 50% of the variation in the observed variable. Research results show that all observed variables are greater than 0.708, of which CC1 is the lowest (outer loading = 0.731). Therefore, all variables in the model are kept for the next step of analysis.

**Table 2.** Quality of observed variables.

|     | AV | CC | CL | CS | DT | IQ | PM | QC | RV | SC |
|-----|------|------|------|------|------|------|------|------|------|------|
| AV1 | 0.870 | | | | | | | | | |
| AV2 | 0.878 | | | | | | | | | |
| AV3 | 0.874 | | | | | | | | | |
| AV4 | 0.873 | | | | | | | | | |
| AV5 | 0.881 | | | | | | | | | |
| AV6 | 0.868 | | | | | | | | | |
| CC1 | | 0.731 | | | | | | | | |
| CC2 | | 0.831 | | | | | | | | |
| CC3 | | 0.878 | | | | | | | | |
| CC4 | | 0.742 | | | | | | | | |
| CC5 | | 0.812 | | | | | | | | |
| CC6 | | 0.814 | | | | | | | | |
| CC7 | | 0.815 | | | | | | | | |
| CL1 | | | 0.819 | | | | | | | |
| CL2 | | | 0.851 | | | | | | | |
| CL3 | | | 0.828 | | | | | | | |
| CL4 | | | 0.818 | | | | | | | |
| CL5 | | | 0.798 | | | | | | | |
| CS1 | | | | 0.913 | | | | | | |
| CS2 | | | | 0.901 | | | | | | |
| CS3 | | | | 0.905 | | | | | | |
| CS4 | | | | 0.925 | | | | | | |
| CS5 | | | | 0.898 | | | | | | |
| DT1 | | | | | 0.837 | | | | | |
| DT2 | | | | | 0.832 | | | | | |
| DT3 | | | | | 0.842 | | | | | |
| DT4 | | | | | 0.832 | | | | | |
| DT5 | | | | | 0.841 | | | | | |
| DT6 | | | | | 0.828 | | | | | |
| DT7 | | | | | 0.865 | | | | | |
| IQ1 | | | | | | 0.869 | | | | |
| IQ2 | | | | | | 0.864 | | | | |
| IQ3 | | | | | | 0.852 | | | | |
| IQ4 | | | | | | 0.867 | | | | |
| IQ5 | | | | | | 0.837 | | | | |

**Table 2.** *Cont.*

|      | AV | CC | CL | CS | DT | IQ | PM | QC | RV | SC |
|------|----|----|----|----|----|----|----|----|----|----|
| PM1  |    |    |    |    |    |    | 0.860 |    |    |    |
| PM2  |    |    |    |    |    |    | 0.841 |    |    |    |
| PM3  |    |    |    |    |    |    | 0.855 |    |    |    |
| PM4  |    |    |    |    |    |    | 0.825 |    |    |    |
| PM5  |    |    |    |    |    |    | 0.830 |    |    |    |
| QC1  |    |    |    |    |    |    |    | 0.827 |    |    |
| QC2  |    |    |    |    |    |    |    | 0.834 |    |    |
| QC3  |    |    |    |    |    |    |    | 0.821 |    |    |
| QC4  |    |    |    |    |    |    |    | 0.825 |    |    |
| QC5  |    |    |    |    |    |    |    | 0.770 |    |    |
| QC6  |    |    |    |    |    |    |    | 0.784 |    |    |
| QC7  |    |    |    |    |    |    |    | 0.780 |    |    |
| RV1  |    |    |    |    |    |    |    |    | 0.841 |    |
| RV2  |    |    |    |    |    |    |    |    | 0.840 |    |
| RV3  |    |    |    |    |    |    |    |    | 0.832 |    |
| RV4  |    |    |    |    |    |    |    |    | 0.822 |    |
| RV5  |    |    |    |    |    |    |    |    | 0.830 |    |
| RV6  |    |    |    |    |    |    |    |    | 0.843 |    |
| RV7  |    |    |    |    |    |    |    |    | 0.835 |    |
| SC1  |    |    |    |    |    |    |    |    |    | 0.819 |
| SC2  |    |    |    |    |    |    |    |    |    | 0.845 |
| SC3  |    |    |    |    |    |    |    |    |    | 0.854 |
| SC4  |    |    |    |    |    |    |    |    |    | 0.815 |
| SC5  |    |    |    |    |    |    |    |    |    | 0.765 |

### 5.2.2. Reliability of the Scale and Convergence

The overall reliability index is greater than or equal to 0.7, proving that the scale is reliable [48,49]. The research results also show that no factor has a composite reliability index less than 0.7 and ranges from 0.911 (SC) to 0.959 (CS); besides, Cronbach's Alpha reliability is greater than 0.8, which is equivalent to good reliability. Thus, the factors of the scale have the appropriate reliability to conduct the next test step.

We used the AVE value to evaluate the convergence of each factor in the scale. AVE > 0.5 means that all parent latent variables explain at least 50% of the variation in each observed child variable [50]. In this model, the AVE values are all greater than 0.5, which means that the scales are all convergent. A detailed convergent validity and reliability is presented in Table 3.

**Table 3.** Convergent validity and reliability.

| Variable | Cronbach's Alpha | rho_A | Aggregate Reliability | Average Extracted Variance (AVE) |
|----------|------------------|-------|-----------------------|----------------------------------|
| Availability (AV) | 0.938 | 0.939 | 0.951 | 0.764 |
| Customer care (CC) | 0.914 | 1.045 | 0.928 | 0.648 |
| Customer loyalty (CL) | 0.881 | 0.885 | 0.913 | 0.677 |
| Customer satisfaction (CS) | 0.947 | 0.948 | 0.959 | 0.825 |
| Delivery time (DT) | 0.93 | 0.931 | 0.944 | 0.705 |
| Information quality (IQ) | 0.910 | 0.911 | 0.933 | 0.736 |
| Payment method (PM) | 0.898 | 0.898 | 0.924 | 0.709 |
| Quality and condition (QC) | 0.910 | 0.91 | 0.928 | 0.65 |
| Reverse logistics (RV) | 0.927 | 0.929 | 0.941 | 0.696 |
| Shipping costs (SC) | 0.878 | 0.881 | 0.911 | 0.673 |

### 5.2.3. Discrimination

Discriminability is ensured when the square root of the AVE for each latent variable is more than the correlation between the latent variables [18]. Therefore, the analysis results show that the values of all square roots of 10 latent variables are higher than the correlation values of latent variables with each other (Table 4). At the same time, the results of HTMT of latent variables are all less than 0.85 [51]. Thus, there is a difference between the structure of the factors in the model.

**Table 4.** Discriminant validity.

|      | AV    | CC     | CL    | CS    | DT    | IQ    | PM    | QC    | RV    | SC    |
| ---- | ----- | ------ | ----- | ----- | ----- | ----- | ----- | ----- | ----- | ----- |
| AV   | 0.874 |        |       |       |       |       |       |       |       |       |
| CC   | 0.098 | 0.805  |       |       |       |       |       |       |       |       |
| CL   | 0.574 | 0.069  | 0.823 |       |       |       |       |       |       |       |
| CS   | 0.576 | 0.150  | 0.507 | 0.909 |       |       |       |       |       |       |
| DT   | 0.535 | 0.129  | 0.416 | 0.644 | 0.840 |       |       |       |       |       |
| IQ   | 0.549 | 0.078  | 0.528 | 0.602 | 0.406 | 0.858 |       |       |       |       |
| PM   | 0.614 | 0.067  | 0.557 | 0.495 | 0.435 | 0.515 | 0.842 |       |       |       |
| QC   | 0.695 | 0.122  | 0.533 | 0.630 | 0.528 | 0.648 | 0.605 | 0.806 |       |       |
| RV   | 0.260 | −0.022 | 0.321 | 0.049 | 0.216 | 0.184 | 0.267 | 0.233 | 0.835 |       |
| SC   | 0.531 | 0.041  | 0.505 | 0.541 | 0.492 | 0.503 | 0.606 | 0.606 | 0.362 | 0.820 |

### 5.3. Analysis of the Linear Structural Model

#### 5.3.1. Multicollinear Assessment

Table 5 indicates the variance inflation factor. Follow that, the VIF values of the research model are all less than 3, showing that this model does not appear to have multicollinearity [52].

**Table 5.** Variance Inflation Factor (VIF).

|      | CL    | CS    |
| ---- | ----- | ----- |
| AV   | 2.376 | 2.351 |
| CC   | 1.033 | 1.028 |
| CL   |       |       |
| CS   | 2.650 |       |
| DT   | 1.944 | 1.586 |
| IQ   | 2.019 | 1.847 |
| PM   | 2.041 | 2.040 |
| QC   | 2.812 | 2.760 |
| RV   | 1.266 | 1.166 |
| SC   | 2.131 | 2.067 |

#### 5.3.2. Relationship Impact Assessment

The authors carried out the bootstrap test with 5000 samples. It is shown that these estimates are reliable in Table 6.

The authors used the bootstrapping technique to evaluate the intermediate relationship between e-logistics service quality and loyalty of Generation Z customers through customer satisfaction. Research results show a significant indirect relationship between QC, IQ, RV, DT, SC, and CL. The highest standardized regression coefficient (O = 0.06) of DT, the lowest QC (O = 0.023), and the normalization coefficient of RV < 0 show that the sign of the regression coefficient from RV to CS carries a bearing. Negative signs or signs of regression coefficient from CS to CL have a negative sign. The results of hypothesis testing are presented in detail in Table 7.

**Table 6.** Bootstrapping analysis results.

| | Original Sample (O) | Sample Mean (M) | Standard Deviation (STDEV) | T Statistics (│O/STDEV│) | *p* Values |
|---|---|---|---|---|---|
| AV → CS → CL | 0.016 | 0.016 | 0.011 | 1.509 | 0.131 |
| QC → CS → CL | 0.023 | 0.023 | 0.011 | 2.033 | 0.042 |
| IQ → CS → CL | 0.042 | 0.042 | 0.016 | 2.634 | 0.008 |
| RV → CS → CL | −0.032 | −0.032 | 0.012 | 2.679 | 0.007 |
| PM → CS → CL | 0.002 | 0.002 | 0.007 | 0.314 | 0.754 |
| DT → CS → CL | 0.060 | 0.060 | 0.022 | 2.782 | 0.005 |
| CC → CS → CL | 0.007 | 0.008 | 0.005 | 1.342 | 0.180 |
| SC → CS → CL | 0.026 | 0.025 | 0.011 | 2.295 | 0.022 |

**Table 7.** Testing of hypotheses.

| | Hypothesis | Original Sample (O) | T Statistics (│O/STDEV│) | *p* Values | Supported |
|---|---|---|---|---|---|
| CS → CL | H1 | 0.164 | 2.944 | 0.003 | Supported |
| DT → CS | H2a | 0.368 | 9.765 | 0.000 | Supported |
| DT → CL | H2b | −0.012 | 0.257 | 0.797 | Not Supported |
| AV → CS | H3a | 0.097 | 1.905 | 0.057 | Not Supported |
| AV → CL | H3b | 0.207 | 3.598 | 0.000 | Supported |
| IQ → CS | H4a | 0.255 | 6.542 | 0.000 | Supported |
| IQ → CL | H4b | 0.164 | 3.024 | 0.003 | Supported |
| QC → CS | H5a | 0.140 | 2.659 | 0.008 | Supported |
| QC → CL | H5b | −0.003 | 0.043 | 0.966 | Not Supported |
| RV → CS | H6a | −0.194 | 6.262 | 0.000 | Supported |
| RV → CL | H6b | 0.161 | 4.603 | 0.000 | Supported |
| CC → CS | H7a | 0.044 | 1.595 | 0.111 | Not Supported |
| CC → CL | H7b | 0.002 | 0.061 | 0.951 | Not Supported |
| SC → CS | H8a | 0.155 | 3.661 | 0.000 | Supported |
| SC → CL | H8b | 0.055 | 1.087 | 0.277 | Not Supported |
| PM → CS | H9a | 0.014 | 0.337 | 0.736 | Not Supported |
| PM → CL | H9b | 0.194 | 3.882 | 0.000 | Supported |

For the effect between the latent variables to be statistically significant, the *p* values must be less than 0.05; however, the results show that the seen hypotheses H2b, H3a, H5b, H7a, H7b, H8b and H9a are not supported due to having *p* value > 0.05.

Meanwhile, the remaining 10 hypotheses are supported; specifically, the influence of DT on CS is the largest (O = 0.368; t = 9.765; *p* = 0.000) and the second biggest factor affecting CS is IQ (O = 0.255; t = 6.542; *p* = 0.039). The next major influence level is SC and QC, respectively. In contrast, RV has the opposite effect with CS (O = −0.194; t = 6.262; *p* = 0.000). For the dependent variable CL, the analysis results show that AV is the most influential factor (O = 0.207; t = 3.598; *p* = 0.000), the second is the PM factor (O = 0.194; t = 3.882; *p* = 0.000), the next most influential level is IQ, CS and the weakest is RV (O = 0.161; t = 4.603; *p* = 0.000).

5.3.3. Explanatory Level of the Independent Variable for the Dependent Variable

The results $R^2$ and $R^2$ adjusted (Table 8), that the independent variables in the model explain 61.7% of the variation in the dependent variable "Satisfaction of Generation Z customers". Meanwhile, the dependent variable "loyalty of Generation Z customers" is explained by 46.1% thanks to the remaining variables in the model.

**Table 8.** $R^2$ and $R^2$ adjusted.

| | $R^2$ | $R^2$ Adjusted |
|---|---|---|
| CL | 0.470 | 0.461 |
| CS | 0.623 | 0.617 |

### 5.3.4. Effect Size $f^2$

In [53] proposed the $f^2$ index table to evaluate the importance of independent variables as follows:

- $f^2 < 0.02$: the effect is extremely small or has no effect.
- $0.02 \leq f^2 < 0.15$: small impact.
- $0.15 \leq f^2 < 0.35$: medium impact.
- $f^2 \geq 0.35$: high impact.

Only DT has a moderate impact on CS, and most of the remaining variables have a small impact or no impact. Table 9 shows the effect size $f^2$ of each independent variable on the two dependent variables.

**Table 9.** Effect size $f^2$.

|  | CL | Impact | CS | Impact |
|---|---|---|---|---|
| AV | 0.034 | Small | 0.011 | Extremely small/No |
| CC | 0.000 | Extremely small/No | 0.005 | Extremely small/No |
| CL |  |  |  |  |
| CS | 0.019 | Extremely small/No |  |  |
| DT | 0.000 | Extremely small/No | 0.226 | Medium |
| IQ | 0.025 | Small | 0.093 | Small |
| PM | 0.035 | Small | 0.000 | Extremely small/No |
| QC | 0.000 | Extremely small/No | 0.019 | Extremely small/No |
| RV | 0.038 | Small | 0.085 | Small |
| SC | 0.003 | Extremely small/No | 0.031 | Small |

### 5.3.5. Evaluation of Out-Of-Sample Predictive Power $Q^2$

In [53] gives the levels of $Q^2$ corresponding to the predictive power of the model as follows:

- $0 < Q^2 \leq 0.25$: low forecast accuracy.
- $0.25 < Q^2 \leq 0.5$: average forecast accuracy.
- $Q^2 > 0.5$: high level of forecast accuracy.

Results presented in Table 10 show that, the corresponding component model of the dependent variable CS has $Q^2 = 0.308$ so this model has average predictive accuracy. Meanwhile, the corresponding component model of the dependent variable CL with $Q^2 = 0.509$ has high predictive accuracy.

**Table 10.** Out-of-sample predictive power $Q^2$.

|  | SSO | SSE | $Q^2$ (=1 − SSE/SSO) |
|---|---|---|---|
| AV | 3060.000 | 3060.000 |  |
| CC | 3570.000 | 3570.000 |  |
| CL | 2550.000 | 1765.551 | 0.308 |
| CS | 2550.000 | 1251.947 | 0.509 |
| DT | 3570.000 | 3570.000 |  |
| IQ | 2550.000 | 2550.000 |  |
| PM | 2550.000 | 2550.000 |  |
| QC | 3570.000 | 3570.000 |  |
| RV | 3570.000 | 3570.000 |  |
| SC | 2550.000 | 2550.000 |  |

## 6. Research Findings and Implications

### 6.1. Findings

#### 6.1.1. Developing e-LSQ Scale

Firstly, delivery time positively affects customer satisfaction of Generation Z online customers, which is also consistent with the research results before [4,35,54,55] that fast

delivery time is the most important factor in customer satisfaction and plays a decisive role in conquering customer satisfaction when shopping online. However, delivery time only indirectly affects customer loyalty through Generation Z customer satisfaction but does not directly impact customer loyalty. While the indirect relationship between delivery time and loyalty of Generation Z online customers is similar to the research results of previous studies [22], only the study in the context of Vietnam's COVID-19 pandemic suggested that delivery time does not directly affect the loyalty of the customer [56]. For further explanation, the authors claimed that as there are now many shipping companies involved in the delivery process, the competition is relatively large, causing these shipping companies to try to optimize the delivery time to retain customers. Furthermore, famous e-commerce platforms in Vietnam have policies such as fast and super-fast delivery for customers who urgently need their products. From the above factors, the difference in delivery time between different sellers is not apparent and not impressive enough for customers to feel the need to be loyal to the seller.

Secondly, product availability has a positive effect on the loyalty of Generation Z e-customers, consistent with the research results [47]. Being out of stock and waiting too long will create opportunities for customers to find competitors, thereby reducing the likelihood of returning, directly affecting the loyalty of online shopping customers.

Thirdly, and interestingly enough, customer care does not affect customer satisfaction and loyalty of Generation Z. This goes against the results of previous studies that customer care is an essential factor for customer satisfaction [39]. However, the authors can explain that this difference comes from the research object is Generation Z customers. The shopping and consumption habits of this generation have changed compared to before. With a huge amount of online information, they can actively search for information and make their own decisions. In addition, previous buyer reviews and famous or influential people's opinions are also crucial for customers to make purchasing decisions. Therefore, sellers need to pay attention to the quality of information, which also positively affects customer satisfaction and loyalty. Product information plays a pivotal role in online purchasing decisions, and it is positively associated with customer satisfaction [57]. The success of the online seller largely depends on the accuracy of the information provided and the product received. Moreover, instead of focusing too much on customer care costs, sellers need to pay more attention to advertising in many ways to build buyers' trust.

Fourthly, payment methods positively affect customer loyalty. This result proves that convenience, speed, and ease in the payment process play an important role in customer retention, especially Generation Z customers, who often use no cash payment methods.

Intense shipping competition and fast delivery policies from major Vietnam e-commerce platforms minimize time differences between sellers. This makes delivery speed insufficient for customer loyalty to a specific seller. Therefore, reverse logistics helps sellers build trust and a pristine image in the eyes of customers. This result is similar to the research results of [56] in China, online shopping customer satisfaction is not affected by reverse logistics while they concluded the opposite result in Taiwan, explaining that there is a difference in the return policies of e-commerce in China and Taiwan, and the rigid concept of profit in China prevents sellers from restricting activities that generate expense. This also happens in the study [55] on satisfaction and loyalty in the e-commerce environment of Malaysia and Qatar.

6.1.2. The Difference in Perception of e-LSQ in Different Demographic Groups

There are differences in almost all aspects of e-LSQ assessment among different gender and income groups. Therefore, sellers need to personalize their products and services, especially e-logistics services, to increase customer satisfaction and loyalty.

As such, this study aims to build a complete scale for e-logistics service quality in terms of delivery time, availability, information quality, product quality and condition, reverse logistics, customer care, shipping costs, and payment methods that affect the satisfaction and loyalty of Generation Z customers. In general, only the customer care factor is not

supported. Regarding customer satisfaction and care, the remaining seven factors all affect the satisfaction and loyalty of Generation Z customers.

*6.2. Implications*

In terms of academics, this study builds a full-scale set of criteria to assess e-logistics service quality that affects customer satisfaction and loyalty of Generation Z consumers. This scale has added several observed variables and developed a new factor called "Payment method" to follow the trend of online shopping and match the research object of Generation Z customers. At the same time, it is indicated that there is a difference in e-logistics service quality between different genders and different income groups in Generation Z.

In practical terms, the results of this study provide several important practical implications for online sellers targeting Generation Z to improve their logistics service quality. Specifically, sellers should focus on the key evaluation criteria identified, including timeliness, accessibility, order accuracy, order condition, order discrepancy handling, and personalized service. To improve on these aspects, they can implement solutions like streamlining warehouse and fulfillment processes, expanding pickup and dropoff options, checking for accuracy before delivery, using automation and barcode scanning, improving packaging, having robust reporting and quick resolution of order issues, and providing personalized services. Additionally, leveraging modern technologies, social media and communication channels favored by Generation Z can also help increase customer trust and satisfaction. Improving logistics service quality in this way will create a competitive advantage and boost the ability to retain Generation Z customers long-term for online sellers.

## 7. Research Limitations and Future Research

Although great efforts have been made to complete the research paper, errors cannot be avoided. Therefore, the study still has some limitations, as follows:

Firstly, the e-LSQ evaluation scale is based on the general assessment of customers for all industries and business types. However, each industry or business type will have certain differences in e-LSQ management. Therefore, this can be the basis for the following studies to develop the e-LSQ scale suitable for their business characteristics.

Secondly, due to time and resource constraints, the sample size is still relatively small, with only 510 survey participants, and concentrated mainly in Hanoi city, where logistics services are more developed than in the rural area. This can lead to different customer perceptions in different regions. The following studies can expand the scope of research to evaluate these factors in the future more specifically.

**Author Contributions:** Conceptualization, A.D.D.; methodology, P.T.B. and H.T.L.; validation, P.T.B., N.T.D. and Q.T.D.; writing—original draft preparation, A.D.D., V.L.T. and P.T.B.; writing—review and editing, A.D.D. and V.L.T. All authors have read and agreed to the published version of the manuscript.

**Funding:** This research received no external funding.

**Institutional Review Board Statement:** Not applicable.

**Informed Consent Statement:** Informed consent was obtained from all the participants.

**Data Availability Statement:** Not applicable.

**Conflicts of Interest:** The authors declare no conflict of interest.

## Appendix A. List of Constructs and Items

| Dimensions | Encode | Items | Source |
|---|---|---|---|
| Delivery time—DT | DT1 | I receive the product in the shortest amount of time from the time of placing the order. | [14,35] |

| | | | | |
|---|---|---|---|---|
| | DT2 | I received the product on time as expected. | [14,35,39] |
| | DT3 | I was supported by the seller by preparing and delivering the goods in the shortest amount of time. | [14,35,39] |
| | DT4 | I still receive the goods as soon as possible in case the goods are not delivered to me on the original schedule. | [14,35,46] |
| Delivery time—DT | DT5 | I was informed by the seller about the expected delivery time. | [37] |
| | DT6 | I am constantly updated by the shipping company on the delivery status of the product. | Authors self-developed |
| | DT 7 | I can choose the shipping company and the type of shipping (fast delivery in 2 h, express delivery). | Authors self-developed |
| | AV1 | I am informed by the seller about the quantity of goods available or will be in stock. | [37,39,47] |
| | AV2 | I was well-informed by the seller regarding the availability of the goods (quantity, color, style...). | [37,39,47] |
| | AV3 | The seller prepares and delivers substitute goods to me in the shortest time in case what I need is temporarily out of stock. | [37,39,47] |
| Availability—AV | AV4 | I am provided with information or suggestions about similar goods in stock by the seller in case the goods I want to buy are temporarily out of stock. | [37,39,47] |
| | AV5 | I have the seller's permission to select the shipment I want. | [37,39,47] |
| | AV6 | The seller provides me with detailed information about the product and when it is available for delivery in the case of pre-ordering. | Authors self-developed |
| | IQ1 | I was provided with full information about the product by the seller. | [14,35] |
| | IQ2 | I can easily find information about the product. | [14,35] |
| Information quality—IQ | IQ3 | I was provided with accurate information about the product by the seller. | [14,35] |
| | IQ4 | I can easily find reviews about the product's quality from previous buyers. | Authors self-developed |
| | IQ5 | I received a timely response from the seller during the purchase. | Authors self-developed |
| | QC1 | I rarely receive goods with damaged packaging. | [13,35] |
| | QC2 | I rarely receive damaged goods due to the seller's fault. | [14,35] |
| | QC3 | I rarely receive damaged goods due to the shipping process. | [14,35] |
| Quality and condition—QC | QC4 | I received the product with the correct specifications as announced by the seller. | [14,35] |
| | QC5 | I received the product in good working condition. | [39] |
| | QC6 | I received the product with the correct model, type, and color according to my order. | Authors self-developed |
| | QC7 | I received all the free-gifts (if any) after buying a product according to the information provided by the seller. | Authors self-developed |
| | RV1 | I can easily choose the channel I want to return goods to the seller. | [34,43] |
| Reverse logistics—RV | RV2 | I don't have to wait long for the seller to recall the product that I want to exchange/return. | [43] |

| | | | |
|---|---|---|---|
| | RV3 | I receive the new product as soon as possible in case I want to exchange it. | [43] |
| | RV4 | I feel that the exchange/return policy is clearly and transparently disclosed by the seller. | [58,59] |
| Reverse logistics—RV | RV5 | I did not have any problems with the shipping company when making an exchange/return. | [58,59] |
| | RV6 | I feel the return and exchange process is streamlined and easy to implement. | Authors self-developed |
| | RV7 | I was enthusiastically supported by the seller during the product return process. | Authors self-developed |
| | CC1 | I feel sympathetic as the seller always tries to "put himself in my shoes". | [13,14] |
| | CC2 | The seller suggested some solutions to the problems I encountered. | [13,14] |
| | CC3 | I feel the seller has enough knowledge and experience to advise me. | [13,14] |
| Customer care—CC | CC4 | I always receive a polite and gentle attitude from the delivery driver. | Authors self-developed |
| | CC5 | I feel the delivery driver always makes an effort to deliver to me whenever a problem arises. | Authors self-developed |
| | CC6 | I get low/free phone charges when I contact customer care service. | Authors self-developed |
| | CC7 | I was enthusiastically advised by the salesperson outside of their business hours. | Authors self-developed |
| | SC1 | I just need to pay the lowest shipping price in the market. | [21] |
| | SC2 | I get to choose from multiple pick-up locations at no extra cost. | [58,59] |
| Shipping costs—SC | SC3 | I do not need to pay for the exchange/return costs (even to the shipping company, the seller,…). | [58,59] |
| | SC4 | I get free shipping from the seller. | [39] |
| | SC5 | I can apply the code to reduce the shipping cost. | Authors self-developed |
| | PM1 | I was given multiple payment options when making a purchase. | Authors self-developed |
| | PM2 | I can transfer money to the delivery driver. | Authors self-developed |
| Payment method—PM | PM3 | I can save time with multiple payment methods to choose from. | Authors self-developed |
| | PM4 | I limit the risks related to cash and proof related to payment. | Authors self-developed |
| | PM5 | My personal information is protected when using electronic payment methods. | Authors self-developed |
| | CS1 | I am satisfied with the quality of the seller's logistics services. | [21,45] |
| | CS2 | I wish there were more sellers with good quality of logistics services. | [21,45] |
| Customer satisfaction—CS | CS3 | I love shopping online at this online store. | [14,46] |
| | CS4 | I am impressed with this seller's logistics service. | [14] |
| | CS5 | I feel the quality from actual experience of service is the same as the expectation. | Authors self-developed |

| | | | |
|---|---|---|---|
| | CL1 | I am more satisfied with the quality of this seller's logistics services than other sellers. | [21,45] |
| | CL2 | I will commit to continuing shopping here. | [21,45] |
| Customer loyalty—CL | CL3 | I will recommend this store's service quality to others. | [14] |
| | CL4 | I am proud to tell others that I bought from this seller. | [37] |
| | CL5 | I will prioritize this seller over other sellers in my next purchases. | Authors self-developed |

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
