# Peer review of "The Impact of the Quality of Logistics Services in E-Commerce on the Satisfaction and Loyalty of Generation Z Customers"

_sustainability, doi:10.3390/su152115294_

Round 1
Reviewer 1 Report
The theme of the paper is certainly interesting, but there are many doubts and aspects that are unclear to me, which go beyond the limits declared on their behalf by the authors, and which make it necessary - again in my opinion - an in-depth revision of the work.
Below, I ask some questions and make some observations.
- For what precise reason did the authors choose Generation Z for their investigation? Do they believe that generation Z is particularly significant for understanding satisfaction and loyalty in the e-commerce sector? and, if so, for what reasons?
- Some research questions seem trivial to me, for example 'H1a: Delivery time has a positive impact on Generation Z customer satisfaction', and 'H1b: Delivery time has a positive impact on Generation Z customer loyalty'. Other questions appear unclear to me, for example 'H8a: Payment method has a positive impact on Generation Z customer satisfaction', and 'H8b: Payment method has a positive impact on Generation Z customer loyalty'. In what sense do the 'payment methods' impact the satisfaction and loyalty of the respondents to the questionnaire? when are they present, when are they easy to use, or what else?
- As regards the sample analysed, is it a statistically representative sample? or, how was the sample selected?
- How were the questionnaires distributed, and how were they collected?
- Regarding the sample, it is written that "In terms of education level, the largest number of people are 324 people who are studying at university (63.5%), followed by the number of people who are postgraduates (18.4%), college students (13.3%) and the lowest is high school graduates (4.7%). Regarding income, there were 251 participants with income from over 10 million to 15 million, accounting for the majority (49.2%), the lowest was those with income below 5 million (3.9%)." So I understand that the majority of the sample is made by students who are currently studying at university. But it is not clear whether the 'postgraduates' and/or the 'college students' and/or the 'high school graduates' are also currently students, or whether they have already acquired the qualification in question. It would seem like this. As for income, the authors then speak generically of 'millions'... of what? I hope for the students that they are dollars, but I imagine they are dongs... But, above all, the question arises: do the students have an income? are they therefore students or workers with a certain qualification in education? or are they student-workers? or is that income that of the family?
In short, in my opinion there are many inaccuracies and aspects to clarify.
Reviewer 2 Report
Thank you for the opportunity of contributing to improve the manuscript entitled "The impact of the Quality of Logistics Services in E-commerce 2 on the Satisfaction and Loyalty of Generation Z Customers", with the focus on determining which factors of e-logistics activities affect the satisfaction and loyalty of Generation Z customers and their influence
Abstract. Well structured. Aim informed, relevance stated, method informed and main conclusions presented.
1. Introduction. The introduction was short but correct. Main studies presented, relevant papers cited, and gap established.
2. Literature Review.
Line 57. Please cite the name of the author(s), but instead of the year of publication, please inform the reference number.
For instance:
From: According to the [10],
To: According to ESCAP [10]...
Lines: 69, 71, 75, 79, 80, 82, 104, 113, 152, 175, 176,188, 201, 204, 216, 218, 224, 238, 239, 240, 301, 379. Please correct reference system.
Line 79. Please exclude the word "defined".
3. Research Hypotheses. Well structured. Hypotheses well justified.
4. Research Methodology.
Line 246, please exclude the colon (:)
Well explained.
5. Research Results.
Well structured.
6. Research Findings and Implications. Well explained.
7. Research Limitations and Future Research. Limitations and future studies informed.
Congratulations to the authors, you did a good job.
Reviewer 3 Report
Dear Authors,
I would like to express my appreciation for your intriguing paper. The topic you've addressed holds significant importance, given the ever-evolving landscape of e-commerce, which continually adapts to meet new market demands and customer expectations. From a scientific perspective, your work appears to be fundamentally sound.
However, I would like to offer some constructive feedback for improvement:
- Introduction: The background in your introduction could be strengthened, and it would be beneficial to clearly articulate your contribution to the field of science.
- Ensure that each table and figure is explicitly referenced in the text to provide clarity to the reader regarding the data you are discussing.
- In Section 5.2.2 and 5.3.3, consider restructuring these subsections by introducing the text before presenting the associated table or figure. This can enhance the flow and comprehension of your content. Comments may be positioned either before or after the table/figure.
- Tables, starting with Table 4 and onwards: Review the spacing between lines to determine if the gaps are excessively large. Reducing these spaces can improve readability.
- It would be beneficial to include a concise section summarizing your conclusions, providing a clear and organized summary of the key findings.
- Regarding your references, I recommend updating them with more recent sources. E-commerce is a rapidly evolving field, and there is a wealth of new and relevant literature available from various countries worldwide.
I hope you find these suggestions helpful in enhancing the quality and professionalism of your paper.
Reviewer 4 Report
This paper focuses on determining which factors of e-logistics activities affect the satisfaction and loyalty of Generation Z customers and their influence, the research has contributed to business enterprises' scale of e-logistics service quality.
This is a well-written paper containing interesting results which merit publication. For the benefit of the reader, however, a number of points need clarifying and certain statements require further justification. There are given below.
1. Several solutions to increase purchases from young people are mentioned in the introduction, what are the solutions?
2. The English-editing needs to be improved and polished for better communication.
3. The text spacing in Tables 4, 7, and 9 is too spaced and should be adjusted.
4. The findings are a bit long, it is recommended to simplify.
5. Line 186-197, please add reference.
6. Formatting of references needs to be harmonized. For example, The first reference has a different year position than the others.
Minor editing of English language required.
Round 2
Reviewer 1 Report
I note positively that the authors have taken my comments into consideration. Many of which, however, were only partially answered in the actual modifications/additions to the article.
In particular, I cannot yet consider the information relating to the sample analyzed to be satisfactory. Just to return to one of the previous observations of mine, it is not clear whether the sample is made up of working students, given that around two thirds of the respondents are university students, but at the same time almost 50% of the total declare an income (which is also found in the middle of the scale): in short, where do these students get the money to make purchases? Is it their own money or the family's? The issue is not trivial: since the research investigates choices about purchasing goods, which obviously involves an outlay of money, it is evidently not the same thing to spend one's own money or take it from the family budget.
In general - and this is certainly not the only case - personally I find situations in which the authors dedicate great attention and show off sophisticated data analysis and processing methodologies to be of little significance, but which are in fact based on small, non-statistically representative samples, relating to limited areas, and so on.
The authors of the present article appear to be aware of some, but not all, of these limitations. And, in my opinion, they should pay more attention to the substance of their investigation, than to the form in which they present it.
